# Effect of NaCl Solution and Simulated Concrete Pore Solution Environment on the Efficiency of Steel Bar Energized Corrosion

**DOI:** 10.3390/ma15197040

**Published:** 2022-10-10

**Authors:** Yang Jianyu, Ye Xin, Jiang Cong, Yang Weijun

**Affiliations:** 1School of Civil Engineering, Changsha University of Science & Technology, Changsha 410114, China; 2China Construction Technology Hunan Co., Ltd, Changsha 410114, China.

**Keywords:** steel bar, energized corrosion, corrosion efficiency, chloride ion, current density

## Abstract

As the main problem of the durability deterioration of concrete structures, the corrosion of steel bars is usually made by the method of electrified corrosion with a short cycle and low cost. However, there is a big difference between the actual corrosion depth and the theoretical corrosion depth after the reinforcement is electrified. In this paper, through the accelerated corrosion test of steel bars, the change law and influence factors of corrosion efficiency of steel bars in concrete simulated pore solution and NaCl solution are studied. The test results show that the corrosion efficiency of reinforcement in the NaCl solution is higher than that in the concrete simulated pore solution, and the corrosion efficiency in the NaCl solution changes in two stages with the corrosion degree of reinforcement. The corrosion efficiency of concrete in the simulated pore solution decreases with the increase of corrosion degree of reinforcement, which is more significant than that in the NaCl solution. Under the same conditions, the corrosion efficiency is higher in the chloride ion solution with high concentration, and the influence of chloride ion concentration change in the simulated pore solution of concrete on the corrosion efficiency is more significant. The corrosion efficiency of reinforcement under low current density is higher than that under high current density.

## 1. Introduction

Reinforced concrete structures are one of the most widely used engineering structures in the world and their service life can reach several decades [1]. The corrosion of reinforcement bars is the major problem in the damage of concrete structures [2,3,4,5], while natural corrosion is a slow and complex process [6]. In order to efficiently observe the corrosion of steel bars, researchers usually accelerate the process by adopting methods of electrified corrosion to develop corroded samples [7,8,9,10,11,12]. According to Faraday’s law, although the amount of corrosion of steel bars can be calculated by galvanic corrosion, most researchers found that the actual corrosion depth of reinforcement is less than the theoretical corrosion depth [13,14,15,16], and some scholars find the opposite conclusion [17,18], which could not guarantee the accuracy and applicability of the results.

Through the rebar corrosion tests under different contexts, the difference between the actual corrosion depth and the theoretical corrosion depth of the rebar accelerated by the electrification cannot be ignored. It has been found that there is a large difference between the actual corrosion depth and the theoretical corrosion depth of the reinforcement in the concrete component when the reinforcement is electrified and corroded in a NaCl solution [19]. The research shows that for the semi-immersion method in which the corroded reinforcement does not directly contact the solution, or the dry wet alternation of a small corrosion current, the actual corrosion degree of the reinforcement is slightly less than the theoretical corrosion depth [20].

In addition, some scholars believe that the reason for the inaccuracy of Faraday’s law of electrolysis is that the current in the circuit does not fully act on the corrosion of the reinforcement, and some of the current participates in other reactions on the anode [21]. They have studied the theoretical value and the actual value of the corrosion of the reinforcement. The ratio of the measured corrosion depth *m* to the theoretical corrosion depth Δ*m* is defined as the corrosion efficiency *α*, which is also known as current efficiency, and can be formulated as Equation (1) [22]:(1)$$\alpha =\frac{m}{\Delta m}\times 100\%$$

An experimental study conducted by Yubun Auyueng [23] indicated that the measured corrosion depth of the reinforcement directly placed in the NaCl solution for galvanic corrosion is aligned with the theoretical results. Other studies have shown that in NaCl solutions, the corrosion efficiency of low current density is higher than that of high current density, and the influence of chloride concentration on the corrosion efficiency of reinforcement is not obvious [24]. Chen Shaojie [25] found that the actual corrosion rate of stirrups is close to the theoretical corrosion rate when the stirrups are fully immersed in a NaCl concentrated solution with different concentrations, and the effect of the electrolyte concentration is small. Since the corrosion efficiency of rebar is affected by diverse factors, it is important to examine the related environmental impacts.

In this study, the effects of a concrete simulated pore solution and NaCl solution on the corrosion efficiency of reinforcing steel by electric current are investigated, and from the degree of corrosion, chloride ion concentration and current density, the changes in corrosion efficiency of reinforcing bars are studied.

## 2. Experimental

### 2.1. Test Sample and Solution

The test sample is made of a HRP400 steel bar with a diameter of 16 mm and a length of 120 mm. Soak the steel bar in hydrochloric acid molarity of 12% for pickling and rust removal, put it into saturated lime water to regenerate the passivation film of the reinforcement, and weigh the initial mass *m*_0_ (accurate to 0.01 g) of the reinforcement after drying. One end of the reinforcement is welded with wires, and both ends of the reinforcement are sealed with waterproof tape. The exposed length of the middle section of the reinforcement is 100 mm, and the corrosion area is about 50.24 cm^2^.

The simulated pore fluid of concrete is a mixed solution of 0.6 mol/L NaOH, 0.2 mol/L KOH and 0.001 mol/L Ca(OH)_2_, with a pH of 13.5; the NaCl solution is selected to control the chloride concentration. The electrified corrosion test scheme is shown in Table 1. X represents the NaCl solution or concrete pore solution; Y represents the corrosion degree of reinforcement.

### 2.2. Corrosion Scheme

The current in the test is calculated by the formula I=i×S. i is the current density and *S* is the corrosion area. The theoretical corrosion depth of the reinforcement Δm is determined by the corrosion rate and the initial mass *m*_0_ of the reinforcement, as shown in Equation (2); then, the estimated energization time of each test sample under different current densities is calculated according to Faraday’s law, as shown in Table 2.
(2)Δm=kIΔt
where *k* is the electrochemical equivalent of metal (g /(A⋅s)), $k_{Fe^{2+}}=2.89\times 10^{-4}g\;/(A\cdot s)$; *I* is the current(A); and Δt is the time(s).

The electrified corrosion accelerating device is composed of a constant current power supply, wires and four electrolytic cells, as shown in Figure 1. The testing process can be described as follows: after connecting No. 4, 5, 6 and 7 steel bars and auxiliary electrodes in series according to the sequence shown in Figure 1, adjust the current to the set value and start energizing. After No. 4, 5 and 6 steel bars reach the predetermined energizing time of 12,000 min, 15,000 min and 18,000 min, respectively, replace with No. 3, 2 and 1 steel bars with the expected energizing time of 9000 min, 6000 min and 3000 min and continue energizing until No. 7 steel bars and No. 1, 2 and 3 steel bars reach 21,000 min at the same time. Turn off the power and end the test.

Take out the reinforcement from the electrolytic cell, acid wash and remove the rust according to the steps mentioned above to obtain the rusted mass *m*_1_, and compare it with the initial mass *m*_0_ to obtain the actual corrosion depth of the reinforcement, and further calculate the corrosion efficiency α, as shown in Equation (3):(3)$$\alpha =\frac{m_{0}-m_{1}}{\Delta m}\times 100\%$$

## 3. Experimental Phenomenon and Result Analysis

### 3.1. Test Phenomenon

After the reinforcement is electrified in the NaCl solution, the solution quickly changes from clear to light green, and the color gradually deepens until it turns to reddish brown. A large number of dark green products are produced around the reinforcement, and most of the corrosion products are deposited at the bottom of the electrolytic cell and gradually become reddish brown deposits. As the test progresses, white substances are produced in the precipitation, which is Fe(OH)_2_, because there is not enough oxygen in the solution to oxidize it into reddish brown Fe(OH)_3_. With the increase of the energization time, gas begins to be generated on the surface of the reinforcement.

When the steel bar is electrified in the simulated pore solution of concrete, a large amount of gas is generated on the surface of the steel bar and the steel bar is not rusted. After that, unevenly distributed rust spots appear on the surface of the steel bar (Figure 2). The gas generated on the surface of the steel bar gradually decreases with the expansion of the rust spots and begins to increase with the test. Most of the corrosion products in the simulated pore solution of concrete are adsorbed on the reinforcement surface because there is a large amount of OH^−^ in the simulated pore solution of concrete, and Fe^2+^ reacts with the nearby OH^−^ to precipitate after entering the solution. Until the end of the test, the reinforcement was not rusted in the simulated pore solution of concrete with chloride ion concentration of 1%, and gas was always generated on the surface of the reinforcement.

The gas generated on the surface of the reinforcement is due to the side reaction in the solution, as shown in Equations (4) and (5). There is an oxygen evolution reaction and chloride ion oxidation in the anode area [22], and the side reaction will consume the current in the circuit, resulting in the difference between the actual corrosion depth of the reinforcement and the theoretical corrosion depth.
(4)$$4{\mathrm{OH}}^{-}-4e^{-}=2H_{2}O+O_{2}↑$$
(5)$$2{\mathrm{Cl}}^{-}-2e^{-}={\mathrm{Cl}}_{2}↑$$

### 3.2. Analysis of Test Results

#### 3.2.1. Corrosion Efficiency of Concrete in Pore Solutions and NaCl Solution

The corrosion efficiency of reinforcement under different current densities and solution environments is shown in Figure 3. It can be found that in the NaCL solution, the corrosion efficiency of the reinforcing steel reached almost 100% on average for 3% concentration of chloride ions, 3 mA/cm^2^ current density of conditions and 5% concentration of chloride ions, 10 mA/cm^2^ current density of conditions; in the concrete simulated pore solution, the highest corrosion efficiency of the reinforcing steel is less than 80%. Therefore, the corrosion efficiency of reinforcement in NaCl solution is higher than that in concrete simulated pore solution. The reason is that the passivation film on the surface of reinforcement is unstable in neutral NaCl solution, and chloride ions in the solution will also damage the passivation film. The reinforcement that loses the protection of the passivation film is oxidized and corroded under the action of current, and the corrosion efficiency is high. However, the passivation film can exist stably in the strongly alkaline concrete simulated pore fluid. Even if the solution contains chloride ions, the passivation film is still in a dynamic process of formation and dissolution, and the passivation film will hinder the corrosion of reinforcement. Another reason is that the content of OH^−^ in concrete pores is high, and the possibility of an OH^−^ reaction under the action of the current is higher. These can cause the low corrosion efficiency.

When the reinforcement is electrified in the simulated pore solution of concrete with a chloride ion concentration of 1%, no corrosion occurs, because in the strong alkaline solution containing a low concentration of chloride ions, a dense and stable passivation film can be formed on the surface of the reinforcement, and the low concentration of chloride ions is not enough to dissolve this passivation film [26]. The intact passivation film separates the Fe atoms in the reinforcement from the solution. The Fe atoms cannot be oxidized into Fe^2+^ under the action of the current and enter the solution. At this time, the OH^−^ in the solution reacts with the reinforcement surface to generate gas, which is consistent with the observed test phenomenon.

#### 3.2.2. Influence of Corrosion Degree of Reinforcement

Figure 3 illustrate that the corrosion efficiency of reinforcement in NaCl solution changes in two stages. In the first stage, the corrosion efficiency is slightly higher than 100% and does not change with the increase of the corrosion degree of reinforcement. In the second stage, the corrosion efficiency is lower than 100% and decreases with the increase of corrosion degree of reinforcement; the corrosion efficiency of reinforcement in the simulated pore fluid of concrete decreases with the increase of the corrosion degree. The effect of the chloride ion on the corrosion efficiency of reinforcement is in two aspects: (1) the OH^−^ generated by the cathodic reaction causes the NaCl solution to be alkaline, and the alkalinity of the concrete simulated pore solution is enhanced. With the test, the alkalinity of the solution becomes stronger and stronger, and the strong alkaline environment will hinder the corrosion of reinforcement; and (2) the rust produced in the solution is adsorbed on the surface of the reinforcement, which hinders the contact between the solution and the reinforcement, resulting in the reduction of the effect of the current. When the corrosion efficiency of reinforcement in the first stage in NaCl solution does not decrease, the passivation film in NaCl solution is difficult to form, and the chloride ion makes it difficult to exert the rust inhibiting effect of the rust, so the corrosion of reinforcement is not hindered.

In Figure 3, the corrosion degree of reinforcement at the turning point of two stages of corrosion efficiency in NaCl solution is not the same. The later the turning point appears in the NaCl solution with low current density and high concentration, the corrosion efficiency can be maintained at the first stage. The corrosion degree of reinforcement at the turning point is shown in Table 3. The corrosion efficiency of the steel bar is slightly higher than 100% because the steel bar is a mixture of iron, carbon, silicon, manganese and other substances. Other substances except iron will not react chemically under the action of current, but will fall off together with the corrosion products, resulting in the actual corrosion depth being greater than the theoretical corrosion depth.

The decline of corrosion efficiency of reinforcement in the NaCl solution and concrete simulated pore solutions is shown in Table 4. It can be seen from the table that the decline of corrosion efficiency in concrete simulated pore solutions is greater than that in the NaCl solution. The decline of corrosion efficiency in both solutions is positively correlated with current density and negatively correlated with chloride ion concentration; the decrease of corrosion efficiency in 3% and 5% NaCl solution is very small, but the difference is large in concrete simulated pore solutions, which indicates that the influence of chloride ion in concrete simulated pore solutions on the decrease of corrosion efficiency is greater than that in NaCl solution.

#### 3.2.3. Influence of Chloride ion Concentration

According to the test results, the influence of the chloride ion concentration change on the corrosion efficiency of reinforcement is analyzed and compared when other conditions are the same. It can be seen from Figure 3 that the corrosion efficiency of reinforcement of the NaCl solution with 5% chloride ion concentration is significantly efficient than the NaCl solution with 3% chloride ion concentration. Therefore, the corrosion efficiency of reinforcement is higher in the chloride ion solution with a high concentration. The influence of the chloride ion on the corrosion efficiency of reinforcement is in two aspects: (1) the passivation film on the surface of reinforcement dissolves and disappears under the action of chloride ion, and the reinforcement is more prone to corrosion after losing the protection of passivation film; and (2) chloride ions will increase the conductivity of the solution, and the resistance in the circuit will reduce the voltage to be applied. The potential of anode reinforcement is lower at a low voltage, and side reactions have difficulty occurring at a low potential.

According to the research in document [22], the influence of chloride concentration change on the corrosion efficiency of reinforcement is not obvious, which is contrary to the conclusion obtained in this test. After comparison and analysis, it is found that the corrosion degree of reinforcement in document [22] is low, and the actual corrosion depth of reinforcement is greater than the theoretical corrosion depth. At this time, the promotion effect of the chloride ion on the corrosion of reinforcement reaches the maximum limit, resulting in that the influence of chloride concentration change on the corrosion efficiency is not obvious. In Figure 3, the corrosion efficiency of reinforcement with a corrosion degree of 5% in the NaCl solution of three concentrations is 102.87%, 103.43% and 103.64%. At this time, the influence of chloride concentration change on corrosion efficiency is not obvious, indicating that when the corrosion degree of reinforcement is low, the influence of chloride concentration change on corrosion efficiency is not obvious. With the increase of the corrosion degree of reinforcement, the influence of chloride concentration change on corrosion efficiency gradually increases.

In Figure 3, the maximum difference in corrosion efficiency between 3% and 5% NaCl solution under four current densities is 1.31%, 7.33%, 7.78% and 13.94%, respectively, while in a concrete simulated pore solution, it is 24.13%, 22.70%, 23.66% and 32.36%, respectively, indicating that the change of chloride ion concentration in a concrete simulated pore solution has a greater impact on corrosion efficiency than in a NaCl solution.

#### 3.2.4. Influence of Current Density

According to the test results, the influence of current density change on the corrosion efficiency of reinforcement under the same conditions is analyzed and compared, as shown in Figure 4. Since the current does not have a corrosion effect on reinforcement in the simulated pore solution of concrete with 1% chloride ion concentration, the corresponding charts are not listed here. It can be seen from Figure 4 that the corrosion efficiency is the highest when the current density is 3 mA/cm^2^, the lowest when the current density is 30 mA/cm^2^, and the corrosion efficiency is between 10 mA/cm^2^ and 20 mA/cm^2^, which indicates that the corrosion efficiency of reinforcement is higher under a low current density. The reasons are as follows: (1) the higher the voltage required by the high current density, the more positive the potential of the reinforcement connected to the positive electrode of the power supply, the higher the possibility of the side reaction under the high potential, and the side reaction consumes the current in the circuit, resulting in low corrosion efficiency; and (2) iron is more active at a high potential and may produce corrosion products with a higher valence. At this time, taking the electrochemical equivalent of Fe^2+^ in Faraday’s law will lead to higher theoretical corrosion depth and lower corrosion efficiency.

## 4. Conclusions

In this study, the corrosion of reinforcing steel in concrete simulated pore liquids and NaCl solutions is investigated and the effects of different levels of reinforcement corrosion, chloride ion concentration and current density on the efficiency of energized corrosion are analyzed. The main conclusions are as follows:The corrosion efficiency of reinforcement in a NaCl solution is higher than that of a concrete simulated pore fluid, and the actual corrosion depth is closer to the theoretical corrosion depth. In the NaCl solution, the corrosion efficiency of the reinforcing steel reached almost 100% on average for the chloride ion concentration of 3%, 3 mA/cm^2^ current density of conditions and the chloride ion concentration of 5%, 10 mA/cm^2^ current density of conditions; in the concrete simulated pore solution, the highest corrosion efficiency of the reinforcing steel is less than 80%.The corrosion efficiency in NaCl solution changes in two stages: in the first stage, the corrosion efficiency does not change with the increase of the corrosion degree of the reinforcement; in the second stage, the corrosion efficiency less than 100% decreases with the increase of the corrosion degree of the reinforcement; the corrosion efficiency of simulated pore fluid of concrete decreases with the increase of the corrosion degree of reinforcement, which is more significant than that in NaCl solution.Under the same conditions, the chloride ion concentration of the 5% NaCl solution of steel corrosion efficiency is significantly higher than the chloride ion concentration of the 3% NaCl solution. The corrosion efficiency of reinforcement is higher in the chloride ion solution with a high concentration. The maximum difference in corrosion efficiency between 3% and 5% NaCl solution under four current densities is 1.31%, 7.33%, 7.78% and 13.94%, respectively, while in concrete simulated pore solution, it is 24.13%, 22.70%, 23.66% and 32.36%, respectively. The change of the chloride ion concentration in the simulated pore solution of concrete has a greater impact on the corrosion efficiency than in the NaCl solution.Under the same conditions, the corrosion efficiency is the highest when the current density is 3 mA/cm^2^, the lowest when the current density is 30 mA/cm^2^, and the corrosion efficiency of 10 mA/cm^2^ current density and 20 mA/cm^2^ current density is between 3 mA/cm^2^ current density and 30 mA/cm^2^ current density. The corrosion efficiency of reinforcement under a low current density is higher than that under a high current density.

In actual reinforced concrete, the influencing factors of the corrosion of reinforcing bars are uncertain. In this study, only experimental studies have been carried out on the chloride ion concentration, current density and the degree of corrosion of reinforcing bars. On the basis of this study, future research could focus on the effects of other practical factors.

## Figures and Tables

**Figure 1 materials-15-07040-f001:**
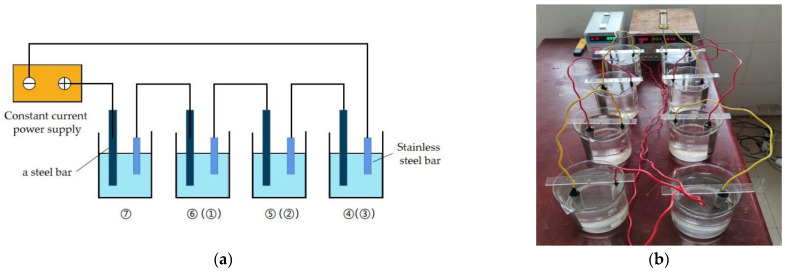
Rebar electrification corrosion device. (**a**) Schematic diagram of corrosion device; (**b**) Corrosion test of steel bar electrification.

**Figure 2 materials-15-07040-f002:**
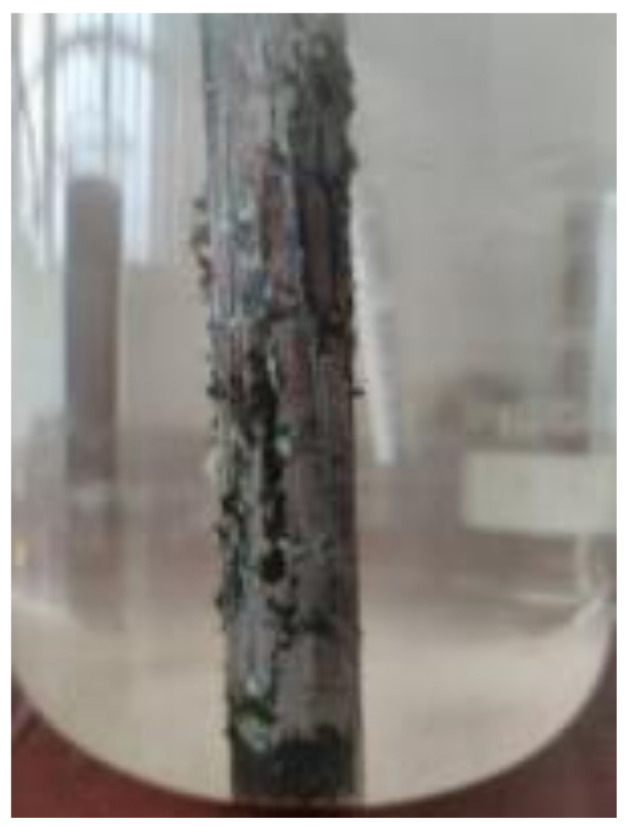
Uneven rust spots on surface of steel bars in simulated pore solutions of concrete.

**Figure 3 materials-15-07040-f003:**
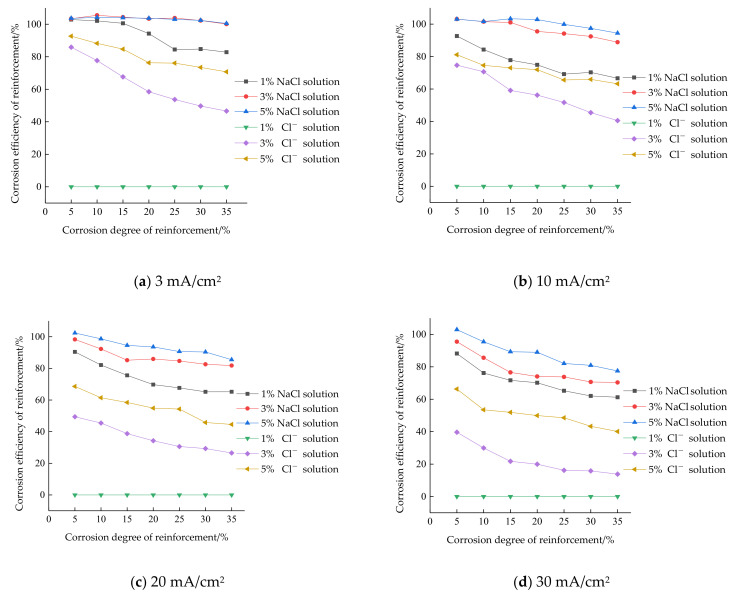
The corrosion efficiency of reinforcement under different current densities and solution environments. (**a**) The corrosion efficiency of reinforcement under 3 mA/cm^2^ current densities; (**b**) The corrosion efficiency of reinforcement under 10 mA/cm^2^ current densities; (**c**) The corrosion efficiency of reinforcement under 20 mA/cm^2^ current densities; (**d**) The corrosion efficiency of reinforcement under 30 mA/cm^2^ current densities.

**Figure 4 materials-15-07040-f004:**
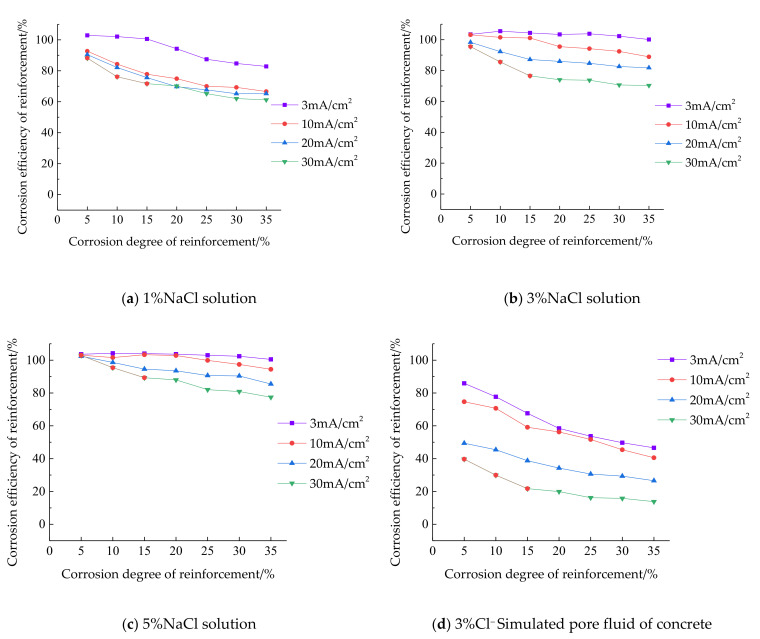
Influence of current density change on corrosion efficiency of reinforcement. (**a**) The corrosion efficiency of reinforcement under 1%NaCl solution; (**b**) The corrosion efficiency of reinforcement under 3%NaCl solution; (**c**) The corrosion efficiency of reinforcement under 5%NaCl solution; (**d**) The corrosion efficiency of reinforcement under 3%Cl− Simulated pore fluid of concrete; (**e**) The corrosion efficiency of reinforcement under 5%Cl− Simulated pore fluid of concrete.

**Table 1 materials-15-07040-t001:** Electrified corrosion test scheme.

No.	Cl^−^ Concentration	Current Density (mA/cm^2^)	Corrosion Degree
X-1-3-Y	1%	3	5%, 10%, 15%,20%, 25%, 30%, 35%
X-1-10-Y		10	
X-1-20-Y		20	
X-1-30-Y		30	
X-3-3-Y	3%	3	5%, 10%, 15%, 20%, 25%, 30%, 35%
X-3-10-Y		10	
X-3-20-Y		20	
X-3-30-Y		30	
X-5-3-Y	5%	3	5%, 10%, 15%, 20%, 25%, 30%, 35%
X-5-10-Y		10	
X-5-20-Y		20	
X-5-30-Y		30	

**Table 2 materials-15-07040-t002:** Estimated power on time of each test sample.

Current Density(mA/cm^2^)	No.						
	1	2	3	4	5	6	7
3	3000	6000	9000	12,000	15,000	18,000	21,000
10	900	1800	2700	3600	4500	5400	6300
20	450	900	1350	1800	2250	2700	3150
30	300	600	900	1200	1500	1800	2100

**Table 3 materials-15-07040-t003:** Corrosion degree of steel bars at two-stage transition of corrosion efficiency in NaCl solution.

Solution	Current Density			
	3 mA/cm^2^	10 mA/cm^2^	20 mA/cm^2^	30 mA/cm^2^
1%NaCl	15%	-	-	-
3%NaCl	35%	15%	-	-
5%NaCl	35%	20%	5%	5%

**Table 4 materials-15-07040-t004:** Decrease of corrosion efficiency in NaCl solution and concrete simulated pore solutions.

Solution	Current Density			
	3 mA/cm^2^	10 mA/cm^2^	20 mA/cm^2^	30 mA/cm^2^
1%NaCl	19.49%	28.12%	27.87%	30.58%
3%NaCl	3.23%	13.84%	16.79%	26.36%
5%NaCl	3.06%	8.32%	16.52%	24.75%
3%Cl^−^ solution	45.75%	45.72%	46.26%	65.21%
5%Cl^−^ solution	23.68%	22.07%	35.02%	39.05%

## Data Availability

The data presented in this study are available on request from the corresponding author.

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
