# Peer review of "Effect of NaCl Solution and Simulated Concrete Pore Solution Environment on the Efficiency of Steel Bar Energized Corrosion"

_materials, 2022, doi:10.3390/ma15197040_

Round 1

Reviewer 1 Report

The manuscript presents a study on the corrosion mechanism of steel bar exposed in high NaCl environment, which is an important piece to assess the reliability and identify the lifespan for the steel-concrete structures built in coastal regions. The experiment is properly set with data and results presented in an easy to follow manner. After reading trough the full article, I encourage the authors to improve the manuscript considering:

1. applying a proofreading for the entire article and update the terms as required. e.g. "corrosion amount" should be "corrosion depth"? "more obvious" could be replaced with "more significant" (in Line 16)? "under different environment" should be "under different contexts"? "obviously less" could be "slight" etc.

2. the Results section could be elaborated as a "Concluding Remarks" section with the key findings of this study highlighted and some recommendations for future studies.

3. Line 24, "Reinforcement corrosion is the main cause of concrete structure failure" could be "Corrosion of reinforcement bar is the major issue results in the failure of concrete structure".

4. Line 25, "In order to accelerate the corrosion process, in the rebar corrosion experiment ..." should be "In order to observe the rebar corrosion efficiently, researchers usually accelerate the process by adopting methods of electrified corrosion to develop corroded samples".

5. Line 47, "and some scholars also call it the current efficiency, as shown in Equation (1)" could be "which is also known as current efficiency, and can be formulated as Equation (1)".

6. Line 49, "Yubun Auyueng et al. found in the experiment ..." could be "An experimental study conducted by Yubun Auyueng et al. indicated ..."

7. Line 50, "is close to the theoretical value" could be "is align with the theoretical results".

8. Line 57, "Therefore, it is of great significance to study the corrosion efficiency and its influencing factors" should be "Since the corrosion efficiency of rebar is affected by diverse factors, it is important to exam the related environmental impacts".

9. Line 66, "The test piece is made of HRP400 steel bar with F16 and length of 120mm" should be "The test sample is made of HRP400 steel bar with a diameter of 16mm and a length of 120mm".

10. Line 89, "The experimental sequence is as follows" could be "The testing process can be described as follows". Line 157, "It can be seen from Figures 3-6 that" could be "Figures 3-6 illustrate that".

Reviewer 2 Report

In this paper, through the accelerated corrosion test of steel bars, the change law and influence factors of corrosion efficiency of steel bars in concrete simulate pore solution and NaCl solution are studied. Based on the initial review, this paper does not clearly articulate the new contributions from a scientific standpoint. Specifically, while the authors reviewed a relatively large body of literature, I cannot see what is new in this paper versus prior work. Regarding these issues, the reviewer recommends that the manuscript should be rejected.

-I would encourage the authors to present a more compelling argument regarding the originality and relevance of their work relative to the previous research work in a clear, easy-to-understand, and verifiable fashion.

- Abstract: The text must be carefully revised. Some sentences contain mistakes. Avoid using the acronym. Explain it in detail. Discuss what the Research Gaps/Contributions are.

- The English writing of the paper is required to be revisited. Please check the manuscript carefully for typos and grammatical errors. Avoiding split infinitives can help your writing sound more formal. Your sentence is unclear or hard to follow. Consider rephrasing the abstract.

-In a research paper, the introduction section is expected to explain the starting background briefly and, even more importantly, the study's originality (novelty) and relevancy are well established. The introduction part does not have a flow or direction. Proper references need to be used rather than using others. The sentences are half-constructed or incomplete so that the language can be improved.

-Provide a proper reference for the equations. It is well known and available in much literature.

- The author must enrich the references with the latest developments in the field. Some of the recent references can be added.

-Conclusions should be more concrete. They should be summarized to show the conclusions of this study clearly. Proper experimental investigations are needed to prove it.

- The numerical investigation considered is a very simple and easy case. Try to involve complex problems to prove the proposed methodology.

-The list could go on, but the bottom line is that the authors need to rewrite the paper or reconsider the research content before being considered for publication in this journal.

Reviewer 3 Report

The paper should thoroughly be revised for grammatical mistakes. The language style used by the authors is unsuitable for a research paper. The whole paper should thoroughly be revised as per the standard style of technical writing.

For different concrete mixes, concrete pore solutions vary widely in composition and properties, usually containing Na + , K + , Ca 2+ , Mg 2+ , Al 3+ , Si 4+ , Fe 3+ , and other ions in different proportions, with the pH varying between 12.4 and 13.5, and redox potentials between − 377 and 139 mV. How did the authors make sure that the solution used in this investigation is a pore solution?

Line 15-19. Please replace “;” with a full stop.

Introduction, Paragraph 2, Line 33: The paragraph started with the word “And”, which needs to be revised. Also, references should be provided by “many scholars

Line 42: Please add a relevant reference for the following sentence “In addition, some scholars believe”

Equation (1), description for the parameter “m” is not given

Line 49: Reference [13] should be provided after “Yubun Auyueng et al.” rather than at the end of the sentence.

Line 51: Please provide relevant references for “Other studies have shown that”.

Line 54: Reference [14] should be provided after “Chen Shaojie et al. Found” rather at the end of the sentence. Also, the spelling “Found” should be “found”.

Line 66: Φ16 should be written as “16mm diameter”.

Line 67: What does mean by “12% hydrochloric acid”? Is it 12% molarity?

Line 73: As per the statement “The simulated pore fluid of concrete is a mixed solution of 0.6 mol/L NaOH, 0.2 mol/L 73 KOH and 0.001 mol/L Ca(OH)2, with a pH of 13.5; NaCl solution is selected to control 74 chloride concentration.” How did the concentration of pore fluid selected? Is 12% standardized?

In Table 1: How did the test scheme finalize? How did Cl concentration 1 to 5%, Current density 3 to 30 mA/sq. m AND Corrosion degrees 5 to 35% chosen?

Line 78: Please remove the full stop from the caption of Table 1. Also, in the Last column of Table 1, please provide commas not “

Figure 2 is unclear

A comparison of actual and simulated corrosion rates should be in 1 figure for a better understanding of the results.

Round 2

Reviewer 2 Report

Paper has been improved according to the reviewers' comments and can be accepted for the publication.

Reviewer 3 Report

Authors have significantly improved the paper; however, the similarity with the internet resources and research publication is 28%, which is quite high. The similarity excluding <10 words similar is also 25%. Therefore, it is recommended to accept the paper with minor revisions.
